# Wood duck nest survival and duckling recruitment is minimally affected by interspecific brood parasitism from hooded mergansers and black-bellied whistling-ducks

Dylan L. Bakner[1¤]*, Kevin M. Ringelman[1], Larry A. Reynolds[2]

1 Louisiana State University AgCenter, School of Renewable Natural Resources, Baton Rouge, Los Angeles, United States of America, 2 Louisiana Department of Wildlife and Fisheries, Baton Rouge, Los Angeles, United States of America

¤ Current address: Department of Natural Resources Science, University of Rhode Island, Coastal Institute, Kingston, Rhode Island, United States of America

* dylanbakner5567@gmail.com

## Abstract

In the southeastern United States, wood ducks (*Aix sponsa*) have historically experienced interspecific brood parasitism (IBP) primarily from hooded mergansers (*Lophodytes cucullatus*), but the recent northward expansion of black-bellied whistling-ducks (*Dendrocygna autumnalis*) has added a new complexity to these interactions. We monitored nest boxes in Louisiana to evaluate the influence IBP had on wood duck daily nest survival rate (after, DSR) and duckling recruitment. We monitored 1,295 wood duck nests from 2020−2023 and found 112 (8.7%) were parasitized by hooded mergansers and 148 (11.5%) by whistling-ducks. Parasitic egg-laying by hooded mergansers lowered wood duck DSR, while DSR for nests parasitized by whistling-ducks was comparable to clutches containing only wood duck eggs. We considered the wood duck capture histories of 2,465 marked female ducklings and 540 banded adult females to estimate a duckling recruitment probability for the entire study period. We recaptured 50 ducklings as adults; 6 (12.0%) hatched from clutches parasitized by hooded mergansers, 1 (2.0%) from a clutch parasitized by a whistling-duck, and 43 (86.0%) from clutches containing only wood duck eggs. The duckling recruitment probability was 0.039 (95% credible interval = 0.028, 0.051). Nest initiation date had a negative effect on recruitment, wherein most recruits hatched from nests initiated earlier in the season. Given only ~9% of wood duck nests contained hooded merganser eggs, we conclude IBP writ large had no detrimental effect on DSR at a population level. The lower DSR of clutches parasitized by hooded mergansers is potentially linked to a high abundance of early-season parasites that produce "dump nests" and these clutches are often abandoned without being incubated. Despite ongoing parasitism by hooded mergansers and the range expansion of whistling-ducks, wood duck productivity in Louisiana appears to be minimally affected by interspecific brood parasitism.

**Data Availability Statement:** All relevant data for this study are publicly available from the Dryad

repository (https://doi.org/10.5061/dryad.gtht76hvt).

**Funding:** Funding for this project was provided by the Louisiana Department of Wildlife and Fisheries, United States Department of Agriculture, Nemours Wildlife Foundation, National Institute of Food and Agriculture McIntire-Stennis grant LAB94294, Louisiana State University College of Agriculture, Louisiana State University Agricultural Center, United Waterfowlers of Florida, and the Louisiana Ornithological Society. Louisiana Department of Wildlife and Fisheries and Nemours Wildlife Foundation assisted with data collection and study design. The remaining funding sources had no role in study design, data collection and analysis, decision to publish, or preparation of the manuscript.

**Competing interests:** The authors have declared that no competing interests exist.

## Introduction

Avian brood parasitism is a reproductive strategy wherein individuals lay their eggs in the nests of others, enabling them to reduce the costs associated with reproduction [1–3]. Brood parasitism comes in two forms: conspecific brood parasitism (after, CBP) where eggs are laid in the nests of the same species [1, 2], and interspecific brood parasitism (after, IBP) where eggs are laid in the nests of a different species [3]. The costs incurred by the parasitized nest host depend on the developmental life history of the offspring. Parasitism of species with altricial young, which need to be fed, imposes higher costs on the host compared to species with precocial young, which are self-feeding [4]. Consequently, defense mechanisms to avoid or mitigate brood parasitism are more common in species with altricial young, while those with precocial young exhibit fewer defensive adaptations.

The offspring of waterfowl species (i.e., Anseriformes) are precocial, and facultative brood parasitism is commonly observed across these taxa [4, 5]. Brood parasitism has been studied extensively in waterfowl that nest in high densities [6–8], over water [9, 10], or in nest cavities [11–13], as parasites can easily find host nests in these environments [4]. Brood parasitism commonly increases the clutch size of nests in the population, resulting in increased nest abandonment and a reduction in hatch rate [i.e., number of eggs hatching; 10, 11, 14]. Moreover, brood parasitism can alter the composition (i.e., the number of host and parasitic young) of the broods [10,15], which can directly influence duckling survival [10, 16, 17]. The implications of brood parasitism are complex, but the pervasiveness of this reproductive strategy makes it an important consideration for the management of waterfowl populations.

Wood ducks (*Aix sponsa*) are a cavity-nesting species and nest box programs are managed by state and federal agencies, nonprofit organizations, and the general public [18, 19]. The shared goal of these programs is to provide safe nest sites for wood ducks to bolster populations; however, these programs often inadvertently increase rates of brood parasitism when nest boxes are highly visible and established at high densities [14]. CBP is particularly prevalent in nest box populations [20–22], and over 95% of nests are parasitized in some areas [18, 23]. CBP has been studied extensively in wood ducks, and recent research indicates that it has a minimal impact on population growth rate [24]; however, little attention has been given to the potential effects of interspecific brood parasitism.

Range-wide, the most common interspecific brood parasite of wood duck nests is the hooded merganser [*Lophodytes cucullatus*; 25–27] whose range overlaps substantially with wood ducks [19]. Hooded merganser eggshells, which are nearly three times stronger than wood duck eggs [28], frequently cause wood duck eggs to break, resulting in reduced hatch rates [29]. Nevertheless, evaluations aimed at understanding the broader consequences to wood ducks subject to interspecific parasitism are lacking. Such research is needed, especially because black-bellied whistling-ducks (*Dendrocygna atumnalis*; after, whistling-duck) are rapidly expanding their breeding distribution into the southeastern United States [15, 30, 31] and are a potential new parasite of southern wood duck nests [32].

Currently, there is one documented observation of whistling-ducks parasitizing a wood duck nest in south Texas [32]. More recent studies have documented whistling-ducks nesting in South Carolina [31] and Louisiana [15], where the timing of nesting overlaps with wood ducks and hooded mergansers. Wood ducks and hooded mergansers both parasitize whistling-duck nests [15]; therefore, it is likely that wood duck nests are routinely parasitized by whistling-ducks. Such potential has raised concern among managers of wood ducks because whistling-ducks exhibit extraordinary rates of CBP, resulting in some of the largest clutch sizes ever documented in waterfowl. For example, observations in south Texas show CBP was present in 100% of nests in one population [33] and clutch sizes regularly exceeded 50 eggs in

another study [the largest containing 101 eggs; 11]. While CBP is a common aspect of the life history of wood ducks and whistling-ducks, the impact whistling-ducks have on breeding wood duck populations in the southeastern United States is unknown.

Given the presence of hooded mergansers throughout much of the breeding distribution of wood ducks, as well as the expanding range of whistling-ducks throughout the southeastern United States, an evaluation of IBP in wood duck nests and its broader consequences to wood duck populations is needed. Such an assessment requires consideration of vital rates most important to wood duck population growth. Recently, Hepp et al. [24] found nest survival and duckling recruitment accounted for 11.4% and 57.7% of the variation in wood duck population growth rate. Here, we report on wood duck nests that were parasitized by hooded mergansers and whistling-ducks in central Louisiana to quantify the prevalence and effects of IBP in wood duck nests. We used a hierarchical approach to 1) describe the composition of clutches, including the size of parasitized clutches and the number of host eggs; 2) evaluate the influence of IBP on wood duck nest survival; and 3) determine the number of wood duck ducklings recruited from parasitized clutches, while also evaluating other factors influencing recruitment.

## Study species

Wood ducks and hooded mergansers are widely distributed across North America, ranging from southern Canada to northern Mexico [19]. Black-bellied whistling-ducks breed throughout South and Central America, Mexico, and the southeastern United States [15, 19, 31]. Wood ducks in northern latitudes migrate south annually, while southeastern U.S. females are year-round residents, resulting in a breeding season of up to 6 months [19]. Females lay ~11 eggs, with an incubation period of approximately 30 days [19, 34]. Hooded mergansers also migrate south each fall, with southeastern females remaining year-round residents [19]. Their breeding season lasts approximately 75 days, during which they lay 10–13 eggs, and incubation takes ~32 days to complete [19, 35]. Whistling-ducks initiate nests from mid-March into early fall in the southeastern United States [15, 31]. Although their clutch size is hard to determine due to high rates of conspecific brood parasitism [15], previous research suggests a single female lays 9–18 eggs [11], with an incubation period of approximately 28 days [36].

## Methods

### Study area

We oversaw nest boxes established by Louisiana Department of Wildlife and Fisheries (LDWF) in Iberville, Pointe Coupee, Rapides, St. Landry, and St. Martin Parishes. Our study sites were located in Sherburne Wildlife Management Area (after, Sherburne), Thistlethwaite Wildlife Management Area (after, Thistlethwaite), Indian Creek Reservoir, Lake Rodemacher, and Oden Lake. Sherburne is 17,652 ha in size, and owned by LDWF, United States Fish and Wildlife Service, and United States Army Corps of Engineers. Sherburne is located along the Atchafalaya River and is primarily bottomland hardwood forest with backswamps and bayous. Sherburne has two moist soil units located on the eastern side of the property known as "North Farm" and "South Farm," which are managed for migratory waterbirds. Thistlethwaite is 4,492 ha of bottomland hardwood forest and is leased to LDWF by a private individual. Indian Creek is a 1,052 ha reservoir surrounded by Alexander State Forest Wildlife Management Area, which is a mix of loblolly (*Pinus taeda*) and longleaf pine (*Pinus palustris*) and hardwood stands. Oden Lake is a smaller private lake ~6.5 km northeast of Indian Creek, and the perimeter of the lake is lined with residential housing. As part of the Oden Lake study site, we also monitored nest boxes located in a cypress swamp directly north of the lake and west of

highway 165. Lake Rodemacher is 1,189 ha in size and ~3 km west of Boyce, Louisiana; the lake is used as a cooling resource when generating power for The Brame Energy Center that is located on the northwest side of the lake. Across all sites, we monitored ~300 nest boxes annually, and most of the nest box populations we oversaw were ~25 years old. Nest boxes were located over open water and accessed by boats, or off the side of levees and navigated to by all-terrain vehicles or on foot.

## Field methods

We visited nest boxes from February 1–July 31 at approximately 7-day intervals during 2020–2023. We monitored the progression of wood duck nests throughout the laying and incubation stages to assign a vital status during each visit. We considered a nest to be alive during the laying stage when we observed an increase in clutch size from the prior visit, otherwise we considered it failed due to nest abandonment. We considered a nest to be alive during the incubation stage when we observed the incubation of the clutch progress from the prior visit, as determined by egg candling [37], otherwise, we considered it to be abandoned. In addition to nest abandonment, any predation event, severe weather, or observer damage that caused laying or incubation progression to cease resulted in a nest failure. We considered a nest successful if it survived to hatch ≥1 egg.

We assigned all eggs from each wood duck nest a numeric identity (after, ID) written with a permanent marker, and we documented the species of each egg [14, 38]. We determined an egg belonged to a wood duck if it was elliptically shaped and a creamy white color [19]; hooded merganser when spherical and white [39]; and whistling-duck if the egg was elliptically shaped, white in color, and showed a blotchy eggshell pattern when viewed through a candling device [15, 37].We recorded which eggs were present or missing at each visit to nests. Following the termination of a nest, we counted the number of eggs that failed or hatched. We counted eggs that went missing, remained unhatched, or were non-viable as failing; egg membranes were used to get a count of hatched eggs [40]. We used egg data when the clutch was found to determine nest initiation dates by back-calculating to the day when the first egg was laid, assuming a laying rate of one egg per day with no partial clutch losses [41]. We determined the clutch size of each nest as being the maximum number of eggs observed in it across all visits, considering all egg species.

We visited wood duck nests daily once they were within three days from their expected hatch date to capture broods from nest boxes. We randomly divided wood duck ducklings in each brood into two groups and fit the first group with web tags [42] and the other with passive integrated transponders tags [after, PIT tag; 43]. Web tags were placed near the tarsometatarsus end of the phalanges [44] and located on the inner webbing of the right foot. We inserted 2x12 mm PIT tags (www.cyntag.com) under the skin, between the scapula using 12-gauge injector needles [www.biomark.com; 43]. Once PIT tags were inserted under the skin, we used 3M Vetbond Tissue Adhesive to suture the injection point. We used both web tags and PIT tags as part of a separate project testing the differences in survival and recapture rates using these two common marking techniques. For the project duration reported here, we recaptured similar proportions of web and PIT tagged individuals; therefore, we assumed the two marking techniques did not influence our results. We returned all ducklings to nest boxes once they were marked. In addition to marking ducklings, we captured adult female wood ducks from nest boxes during incubation [42]. Once captured we fitted each female with a United States Geological Survey aluminum leg band. Waterfowl species exhibit female-based philopatry, which allowed us to recapture marked female ducklings and adults from nest boxes in

subsequent breeding seasons [42, 45]. We checked all unbanded adults for web tags and used a handheld radio frequency identification device (www.biomark.com) to check for PIT tags.

## Clutch descriptions

We used our egg data to categorize nests into clutch types based on the presence of IBP and nest initiation date. We termed wood duck nests containing ≥1 hooded merganser egg "mixed merganser clutches" and those with ≥1 whistling-duck egg "mixed whistling-duck clutches"; all other nests were "normal clutches," which may have contained parasitic eggs from conspecifics. To facilitate the comparison of parameter estimates between mixed clutches and normal clutches, we divided the normal clutches into two distinct periods that aligned with the temporal periods in which we observed mixed merganser and mixed whistling-duck clutches. We labeled nests initiated before April 1 "early clutches," as parasitic merganser eggs were most likely to be observed in wood duck nests initiated prior to that date at our study sites. Conversely, nests initiated following that period were referred to as "late clutches," when parasitic whistling-duck eggs begin to appear in wood duck nests at our study sites. We confirmed April 1 was a satisfactory cutoff date for assigning early and late clutches by examining the variation in nest initiation dates for mixed clutches through visual plots.

## Clutch type analyses

For each clutch type, we modeleded 1) clutch size, 2) the number of wood duck eggs, and 3) the number of hatched wood duck eggs. To do so, we constructed three Bayesian Poisson models considering clutch type as a fixed effect. We compared the posterior means of early clutches to mixed merganser clutches, and late clutches to mixed whistling-duck clutches. When predicting the number of hatched wood duck eggs, we considered only successful nests.

## Nest survival

We estimated daily nest survival rate (after, DSR) and overall nest success (i.e., hatched ≥1 egg) within a Bayesian framework [46, 47]. To do so, we constructed daily encounter histories for each nest as described by Schmidt et al. [46]. Our encounter histories represented a chronological record of live-dead observations, starting with a 1 on the first day the nest was observed alive. For successful nests, the encounter history contained a continuous series of 1's for each day we observed it. In the case of failed nests, the encounter history included 1's for the days we observed the nest alive, followed by NA values for the days between the last observed alive state and the subsequent visit confirming the nest failed. The use of NA values specified our uncertainty of which day nest failures occurred [48, 49]. The encounter history of failed nests ended with a 0, representing the day field observers first detected the nest failed. We estimated DSR as a series of Bernoulli trials:

$$y_{i,t} \sim Bernoulli(y_{i,t-1} * DSR_i),$$

$$logit(DSR_i) = \alpha + \beta_1 * clutch\ type_i + \beta_2 * clutch\ size_i + \beta_3 * clutch\ size_i^2 + site\ ID,$$

where $y_{i,t}$ is the assigned vital status of nest $i$ at time $t$ and $y_{i,t-1}$ is the assigned vital status of nest $i$ at time $t-1$. We used a logit link function to evaluate the relationship between covariates and DSR, where $\alpha$ is baseline DSR on the logit scale. We calculated overall nest survival by exponentiating DSR to the forty-first power assuming the laying and incubation stages take 11 and 30 days to complete [18, 19]. Our linear predictor included clutch type and clutch size along with its quadratic form as fixed effects and site ID as a random effect. We included the

quadratic term for clutch size as we predicted the effects of excessive brood parasitism could be non-linear, wherein wood ducks may abandon nests at some large clutch size threshold [14]. We tested this by comparing the clutch size of abandoned nests to successful nests. To do so, we constructed another Bayesian Poisson model considering nest fate (i.e., abandoned or successful) as a fixed effect in the model. We compared the posterior clutch size means of abandoned nests to successful nests. In this analysis we excluded any nests that failed due to predation events, severe weather, or observer damage.

## Duckling recruitment

We used our duckling and adult capture-mark-recapture data to estimate a single duckling recruitment probability. We assumed half the marked ducklings from each brood were females and excluded male ducklings from our analysis. Female ducklings were recaptured by hand from nest boxes as adults, where PIT-tagged individuals were detected using handheld RFID readers. We used a Bayesian multistate model to derive estimates of apparent survival ($\Phi$) and capture probability [$\varphi$; 50, 51]. We built individual encounter histories where individuals could be in one of the following three states: duckling, adult, and dead. To execute the model, we first constructed a state-transition matrix to describe the state of individuals at time $t + 1$ given their state at time $t$:

$$
\begin{array}{c}
state_t \\
\end{array}
\begin{array}{cc}
& \begin{array}{ccc} duckling & adult & dead \end{array} \\
\begin{array}{c} duckling \\ adult \\ dead \end{array} &
\left[ \begin{array}{ccc}
0 & \Phi_d & 1 - \Phi_d \\
0 & \Phi_a & 1 - \Phi_a \\
0 & 0 & 1
\end{array} \right],
\end{array}
$$

where the matrix defines the true states and specifies that once duckling $i$ advances to the adult state, it remains there until mortality occurs. We then built an observation matrix to link the true states to what we observed in the field:

$$
\begin{array}{c}
state_t \\
\end{array}
\begin{array}{cc}
& \begin{array}{ccc} \textit{seen as duckling} & \textit{seen as adult} & \textit{not seen} \end{array} \\
\begin{array}{c} duckling \\ adult \\ dead \end{array} &
\left[ \begin{array}{ccc}
1 & 0 & 0 \\
0 & \varphi_a & 1 - \varphi_a \\
0 & 0 & 1
\end{array} \right],
\end{array}
$$

where the matrix describes the observations on individual $i$ at time $t$, given the latent state at time $t$. We detected all individuals observed in the duckling state; therefore, we fixed this parameter at 1. We were only able to estimate $\varphi$ for the adult state because ducklings were not observable beyond their hatch dates. While the multistate model provides estimates of $\Phi_a$ and $\varphi_a$ we do not report them here as they are not needed for our evaluation. We defined recruitment probability as the probability female duckling $i$ marked at time $t$ survives to its breeding age (1 year old) and returns to the study area at time $t + 1$ to be captured as a breeding adult, which is analogous to $\Phi_d$.

We built a separate Bayesian multilevel model to evaluate the effects brood-level variables had on the recruitment probability [$x_i$; 52]. We employed a zero-inflated binomial distribution

to fit our data:

$$x_i \sim Zero-Inflated\,Binomial(n_i, p_i, \pi_i),$$

$$logit(p_i) = \alpha + \beta 1 * nest\,initiation\,date_i + \beta 2 * brood\,size_i + brood\,ID,$$

where we considered the quantity of female wood duck ducklings from brood *i* as the number of trials ($n_i$) and the number later observed in the adult state as successes ($p_i$); $\pi_i$ was the zero-inflated probability. We considered nest initiation date and brood size as fixed effects and brood ID as the random effect in our model. Small sample sizes precluded us from including brood composition (i.e., number of merganser or whistling-duck ducklings in each wood duck brood). We were unable to assign a nest initiation date to 17 nests; the mean nest initiation date replaced these missing values [53].

Prior to running our models, we scaled all variables to have a mean of 0 and a standard deviation of 1. This process places all covariates on the same scale, facilitating the comparison of beta estimates across different covariates [54]. We calculated the percentage of variation explained by our fixed effects to evaluate their contribution to our models [55]. This involved constructing an additional nest survival and multilevel recruitment model containing only the random effect terms. We used the variance estimates ($\hat{\sigma}^2$) obtained from our fixed effects and random-effects-only models, to execute our calculation using the seventh equation from Grosbois et al. [55]:

$$\frac{\hat{\sigma}^2(random\,effects\,only\,model) - \hat{\sigma}^2(fixed\,effects\,model)}{\hat{\sigma}^2(random\,effects\,only\,model)}$$

We fit our DSR and multistate models in JAGS 4.3.1 [56] using the R2jags 0.7–1 package [57] in Program R 4.2.2 [58]. We used Uniform (0, 1) priors for modeling DSR, Normal (0, 0.01) priors for fixed effect coefficients, and Uniform (0, 5) priors to estimate site ID variance ($\sigma^2$). For our recruitment model, we used Uniform (0, 1) priors for estimates of $\Phi$ and $\varphi$. We employed the brms 2.19.0 package [52] to fit our multilevel models, leveraging its interface with Stan [59]. For all multilevel models, we used default vague prior settings offered by the brms package for estimating fixed effect coefficients and random effect variances. For all our models, we ran 4 chains of 30,000 iterations, considered 10,000 as burn-in, and saved every tenth iteration. We confirmed our models converged via inspection of trace plots and by using the Gelman-Rubin statistic, where a $\hat{R}$ value < 1.05 indicated convergence [60]. From our analyses, we report posterior means ($\mu$) and coefficient values ($\beta$), where credible intervals (after, CI) that did not overlap zero indicated significance effects from model coefficients at the 0.05 alpha level. For models containing clutch type as a covariate, normal clutches were used as the reference group.

We conducted our work under U.S. Fish and Wildlife Service banding permit #06669 and Special Use Permit 43614-20-04; Louisiana Department of Wildlife and Fisheries state collecting permits WDP-20-037 and WDP-21-060, and Wildlife Management Area Permit WL-Research-2020-03; Louisiana State University Institutional Animal Care and Use Protocol A2019-27.

## Results

We monitored a total of 1,295 wood duck nests from 2020–2023. We observed mixed merganser clutches initiated from January 26–April 29 and mixed whistling-duck clutches initiated from February 12–June 14. We determined April 1 was a satisfactory cutoff date for assigning early and late clutches (Fig 1). Of the nests we monitored, 112 (8.7%) were mixed merganser

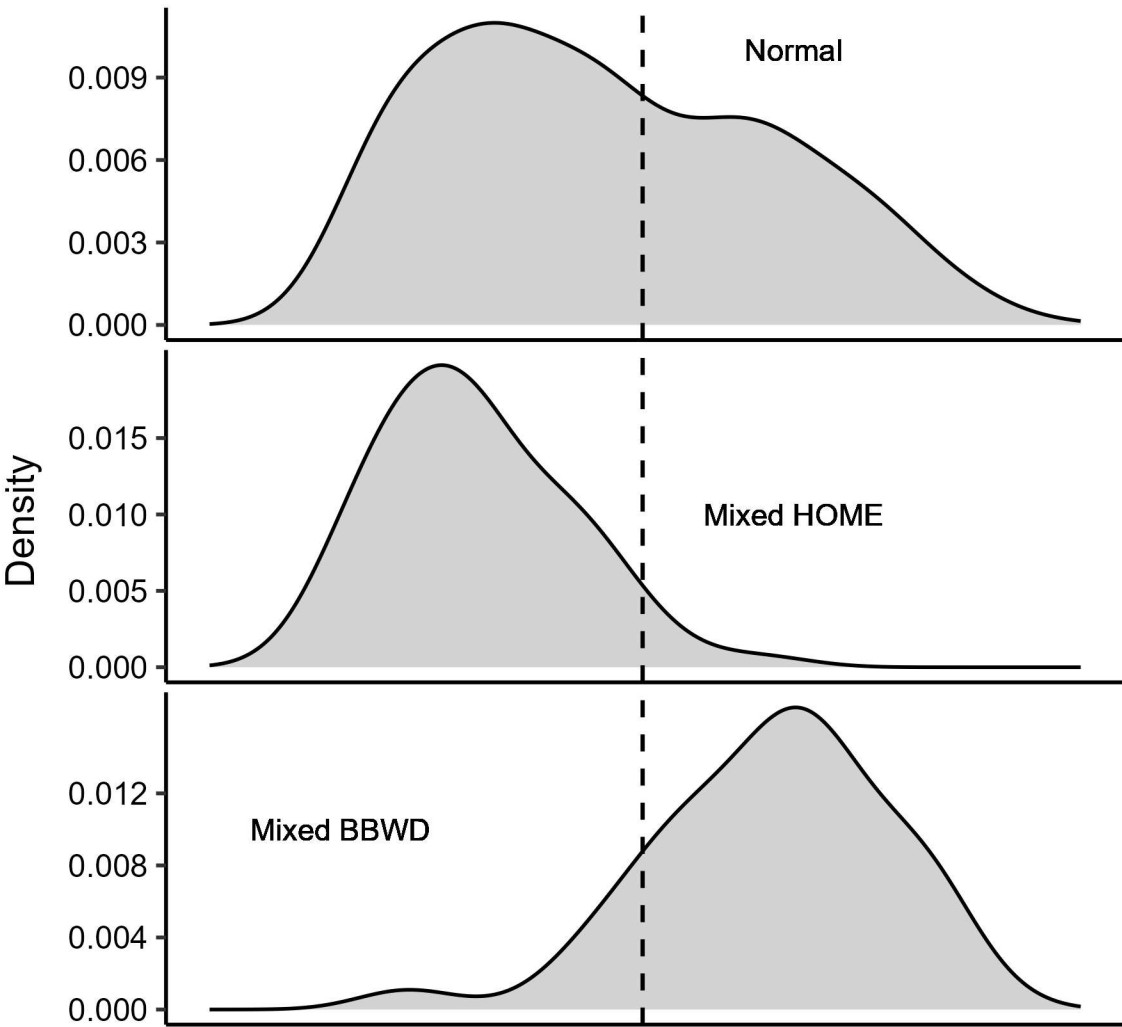

**Fig 1. Density plots showing nest initiation dates for wood duck (Aix sponsa) clutches categorized into three groups: Normal clutches (i.e., clutches with only wood duck eggs), clutches containing parasitic hooded merganser eggs (Lophodytes cucullatus; Mixed HOME),, and clutches containing parasitic black-bellied whistling-duck eggs (Dendrocygna autumnalis; Mixed BBWD).** The vertical dashed line represents April 1, which is used as the cutoff date for early and late wood duck nests.

clutches, 614 (47.5%) were early clutches, 148 (11.5%) were mixed whistling-duck clutches, and 418 (32.4%) were late clutches. A total of 3 wood duck nests contained both parasitic merganser and whistling-duck eggs, which we excluded from our analyses. A total of 46 (41.1%) mixed merganser, 303 (49.3%) early clutches, 82 (55.4%) mixed whistling-duck, and 188 (45.0%) late clutches were successful. We observed a total of 16,454 wood duck eggs, with 1,440 (8.8%) being laid in mixed merganser clutches, 8,777 (53.3%) in early clutches, 1,585 (9.6%) in mixed whistling-duck clutches, and 4,652 (28.3%) in late clutches. A total of 418 (29.0%) wood duck eggs hatched from mixed merganser clutches, 3,578 (40.8%) from early clutches, 607 (38.3%) from mixed whistling-duck clutches, and 1,734 (37.3%) from late clutches. Additionally, mixed clutches hatched 165 hooded mergansers and 196 whistling-ducks. Of the mixed merganser clutches that failed, 20 (17.9%) nests failed due to abandonment, 45 (40.2%) were depredated, and 1 (0.01%) failure was undetermined. Among the early clutches that were unsuccessful, 77 (12.5%) nests were abandoned, 230 (37.5%) were

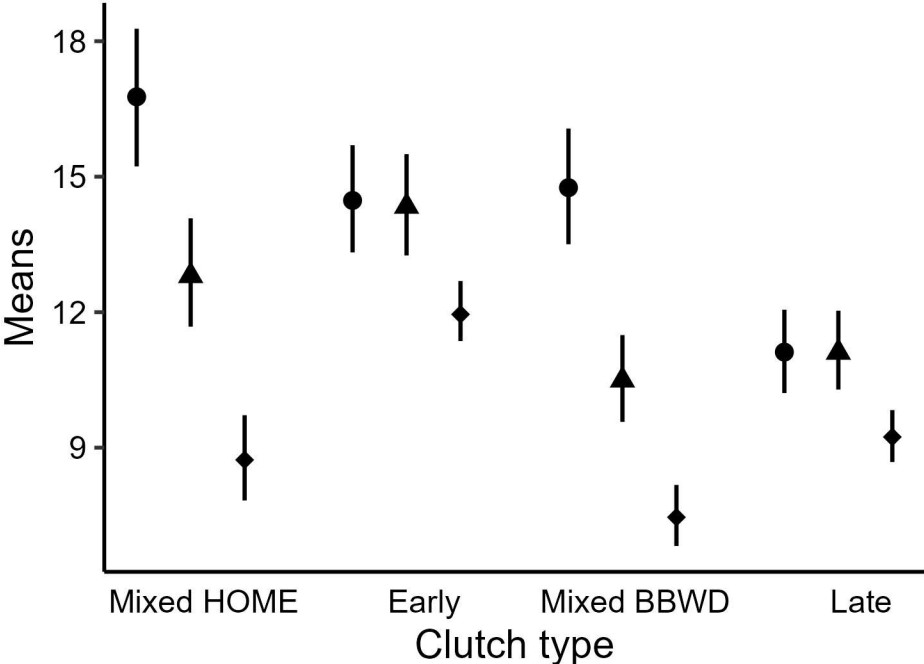

**Fig 2. Posterior means from three Bayesian Poisson models estimating clutch size (i.e., total number of eggs), number of wood duck (Aix sponsa) eggs, and number of hatched wood duck eggs, with clutch type as a 44fixed effect.** Early clutches contained only wood duck eggs and were initiated before April 1. Late clutches contained only wood duck eggs and were initiated after April 1. Mixed HOME clutches were wood duck nests containing ≥ 1 hooded merganser (Lophodytes cucullatus) egg and were primarily observed before April 1. Mixed BBWD clutches were wood duck nests containing ≥ 1 black-bellied whistling-duck (Dendrocygna autumnalis) egg and were primarily observed after April 1.

depredated, and 4 (0.01%) failure causes were undetermined. Of the mixed whistling-duck clutches that failed, 18 (12.2%) nests failed due to abandonment and 48 (32.4%) were depredated. Among the late clutches that were unsuccessful, 55 (13.2%) nests were abandoned, 174 (41.6%) were depredated, and 1 (0.001%) failure was undetermined.

All our models achieved convergence according to trace plot inspections and R̂ values. Considering early clutches as the reference group, mixed merganser clutches were larger (β = 0.15, CI = 0.09, 0.20), late clutches were smaller (β = -0.27, CI = -0.30, -0.23), and mixed whistling duck clutches were comparable (β = 0.02, CI = -0.03, 0.07; Fig 2). There were fewer wood duck eggs in mixed merganser clutches (β = -0.11, CI = -0.18, -0.05), late clutches (β = -0.26, CI = -0.29, -0.22), and mixed whistling duck clutches (β = -0.31, CI = -0.37, -0.26; Fig 2). Likewise, fewer wood duck eggs hatched from mixed merganser clutches (β = -0.32, CI = -0.43, -0.21), late clutches (β = -0.26, CI = -0.32, -0.20), and mixed whistling duck clutches (β = -0.47, CI = -0.56, -0.39; Fig 2).

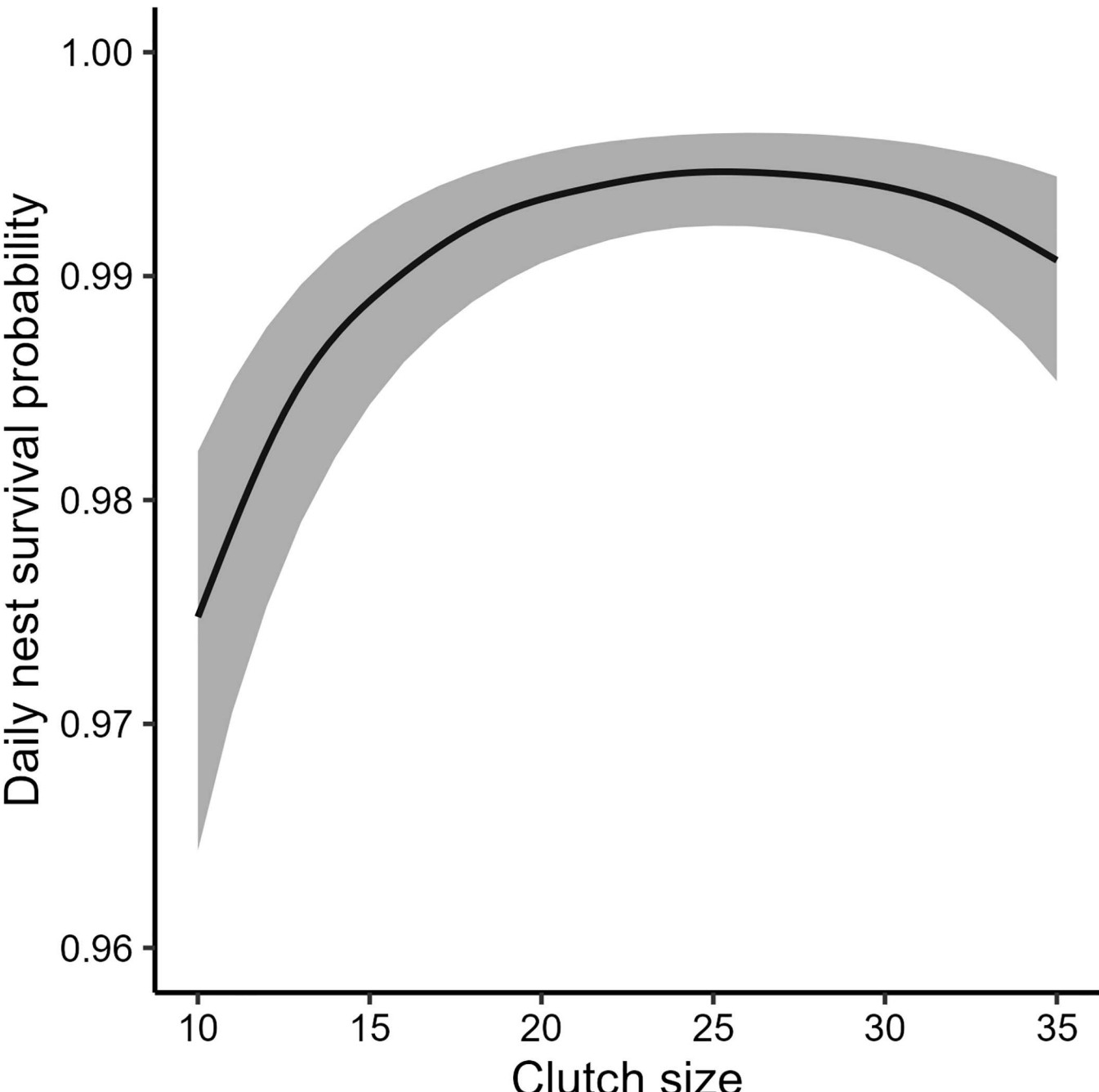

**Fig 3. Daily survival rate of wood duck (Aix sponsa) nests across different clutch sizes.** Clutch size was the maximum number of eggs (i.e., host and parasitic eggs combined) observed in each nest across all visits.

Baseline DSR from our model containing clutch type and clutch size along with its quadratic form as fixed effects was 0.981 (CI = 0.974, 0.987), which yielded a nest success estimate of 0.455 (CI = 0.340, 0.585). The variance estimate for the fixed effects model was 0.603 (CI = 0.552, 0.705) and 0.628 (CI = 0.568, 0.742) for the random-effects-only model; therefore, model covariates explained 3.9% of the variation in DSR across our study sites. Clutch size ($\beta$ = 2.35, CI = 2.12, 2.58) and the quadratic term ($\beta$ = -1.50, CI = -1.69, -1.29; Fig 3) had a positive

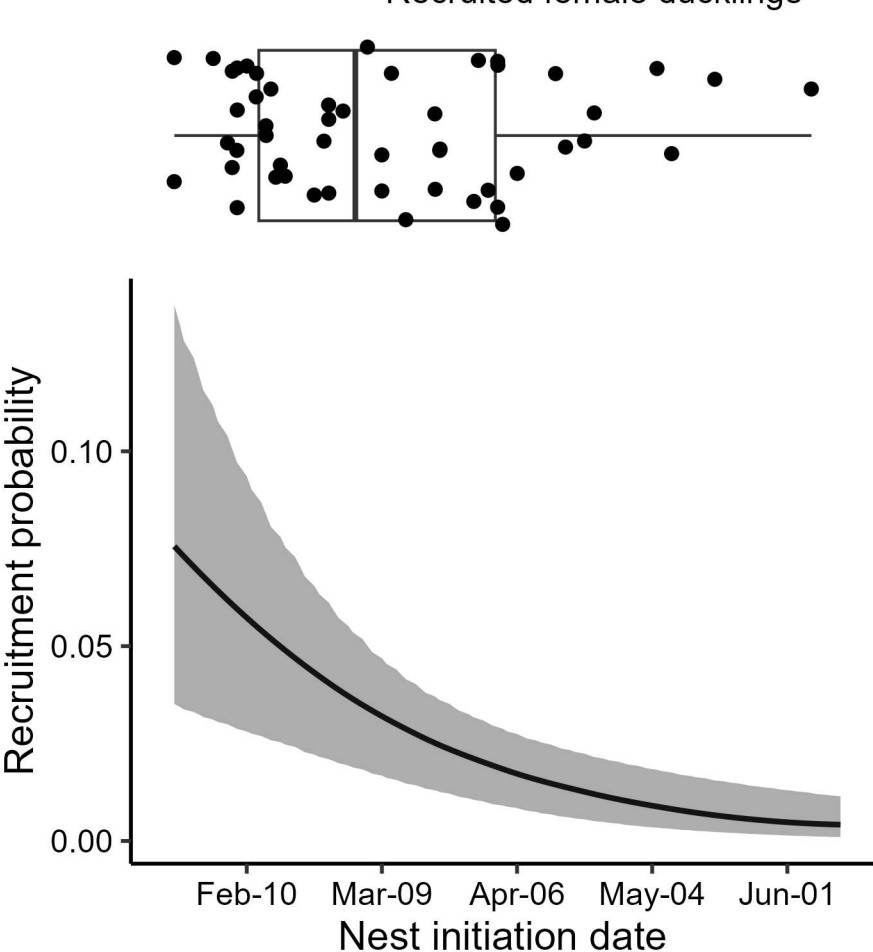

**Fig 4. Recruitment probability for wood duck (Aix sponsa) ducklings across nest initiation dates.** The box plot shows the variation in nest initiation dates for ducklings recaptured as breeding adults. All plots are scaled to the same x-axis.

and negative influence of DSR, respectively. The survival benefits of a large clutch appeared to asymptote at ~23 eggs (Fig 3). Mixed merganser clutches had a negative effect on DSR (β = -0.67, CI = -0.99, -0.56), yielding a DSR estimate of 0.964 (CI = 0.947, 0.977). DSR was similar for all other clutch types and ranged from 0.980–0.983. We found little evidence of larger clutches being abandoned by wood ducks, as abandoned clutches (μ = 12.0, CI = 10.6, 13.7) were smaller than successful ones (μ = 15.5, CI = 13.7, 17.5).

We considered the capture histories of 2,465 female ducklings marked from 480 successful nests and 540 adult females captured from nest boxes. We tagged 186 (7.6%) female wood duck ducklings from 41 mixed merganser clutches, 1,389 (56.4%) from 223 early clutches, 224 (9.1%) from 64 mixed whistling-duck clutches, and 666 (27.0%) from 152 late clutches. We recaptured 50 (2.0%) ducklings as breeding adults with 6 (3.3%) returning from mixed merganser clutches, 30 (2.2%) from early clutches, 1 (0.5%) from a mixed whistling-duck clutch, and 13 (1.2%) from late clutches.

The mean recruitment probability was 0.039 (CI = 0.028, 0.051) across our entire study period. Nest initiation date had a negative effect (β = -0.73, CI = -1.13, -0.35; Fig 4) on duckling

recruitment probability, while brood size had no effect (β = -0.20, CI = -0.54, 0.14). The across-brood variance estimate was 0.544 (CI = 0.025, 1.352) for the model containing fixed effects (i.e., nest initiation date and brood size) and 0.671 (CI = 0.036, 1.527) for the random-effects-only model, suggesting fixed effects explained 18.8% of the across-brood variation in recruitment probability.

## Discussion

We found hooded mergansers and whistling-ducks each parasitized wood duck nests for ~3 months during the breeding season, and wood duck nests were simultaneously parasitized by both species for a period of ~1.5 months (Fig 1). Approximately 20% of the wood duck nests we observed were interspecifically parasitized. Mixed merganser clutches had a lower DSR, whereas DSR for mixed whistling-duck clutches was commensurate to early and late normal clutches. Explaining the lower DSR observed in mixed merganser clutches presents a challenge, but it is potentially linked to a high abundance of early-season hens that are strictly brood parasites that do not incubate. These brood parasites often produce "dump nests" that are never incubated (Semel et al. 1988), which appear in our data as failed (abandoned) nests.

   We found IBP had minimal effects on the DSR of wood duck nests when considering clutch size and type. In contrast, IBP contributed to larger clutch sizes, with DSR increasing up to a clutch size of ~23 eggs, after which a significant quadratic effect caused DSR to decline (Fig 3). The relationship between larger clutch sizes and higher DSR may be driven by partial clutch predation by red-bellied woodpeckers (*Melanerpes carolinus*), which occurred when nest boxes were vacated by wood ducks during laying and incubation recesses [61]. We hypothesize that parasitic egg laying and large clutch sizes (i.e., clutch sizes ≤ 28 eggs) act as a protective mechanism against nest abandonment. Without the addition of eggs from parasitic individuals, wood duck clutches would be depleted by partial predation, particularly during the laying period, which would increase the probability of nest abandonment.

   The mean recruitment probability from our study is comparable to estimates from wood duck populations in South Carolina, which ranged from 2.24–6.84% [42]. Surprisingly, ducklings hatched from mixed merganser clutches had the highest apparent return rate (3.2%); however, most of these recaptures were from a single study site, where the other factors such as brooding habitat may have been favorable. Nest initiation date was the only important predictor of recruitment probability and strongly suggests that earlier-hatched female ducklings have a greater chance of entering the nest box population as breeding adults. Hepp et al. [42] analyzed 6 years of data for wood duck ducklings web-tagged in South Carolina and found no relationship between duckling recruitment and hatch date. Similarly, studies estimating 30-day duckling survival found hatch date had little influence on survival [17, 62]. While our findings contradict these results, it is worth noting that several studies on other waterfowl species have consistently shown higher recruitment rates for ducklings hatched earlier in the spring [63–65]. In our analysis, we found that nearly 75% of the recruited ducklings hatched from nests initiated prior to mid-April. Recruited ducklings were more likely to have hatched from clutches that were initiated around the peak nest initiation date, which typically occurred in early-March, as indicated by the distribution plot in Fig 1. The bimodal distribution in Fig 1 coincides with first-round nesters and renesters [66]; hence, our recruitment analysis suggests that first-round nesters likely recruit more female ducklings. Another explanation for the lower recruitment probability for ducklings hatched in the late spring and summer is the diminished brood habitat, where water levels are lower at many of our study sites either due to drier weather conditions or intentional drawdowns for management purposes (e.g., moist soil and invasive aquatic vegetation management).

Overall, our findings indicate that IBP in wood duck nests has minimal effects on DSR and duckling recruitment probability; however, it is worth exploring potential reductions in individual fecundity as a result of mixed clutches. To evaluate these costs, future research could use modern genetic techniques to measure the reduction in hatched host eggs in mixed clutches, which includes determining the number of eggs belonging to conspecific parasites [67]. Furthermore, it would be valuable to investigate other vital rates during the breeding season, such as breeding propensity and 30-day brood survival, particularly in relation to breeding densities of whistling-ducks. Personnel from Louisiana Department of Wildlife and Fisheries and Louisiana State University Agricultural Center report whistling-ducks displaying aggressive behavior, potentially discouraging wood ducks from accessing nest boxes and even attempting to drown wood duck broods. These observations present promising avenues for future research studies.

## Supporting information

**S1 File.**
(R)

**S2 File.**
(CSV)

**S3 File.**
(CSV)

**S4 File.**
(TXT)

**S5 File.**
(CSV)

**S6 File.**
(CSV)

**S7 File.**
(CSV)

## Acknowledgments

Funding for this project was provided by the Louisiana Department of Wildlife and Fisheries, United States Department of Agriculture, Nemours Wildlife Foundation, National Institute of Food and Agriculture McIntire-Stennis grant LAB94294, Louisiana State University College of Agriculture, Louisiana State University Agricultural Center, United Waterfowlers of Florida, and the Louisiana Ornithological Society. Louisiana Department of Wildlife and Fisheries and Nemours Wildlife Foundation assisted with data collection and study design. The remaining funding sources had no role in study design, data collection and analysis, decision to publish, or preparation of the manuscript. We thank the Louisiana Department of Wildlife and Fisheries for providing field housing and supplying assistance with collecting data. We especially thank C. Jones and T. Vidrine for helping with issues that arose during fieldwork. We thank all the technicians who served on this project: N. Bosco, D. Ducote, J. Dubman, A. Jackson, N. Ragucci, C. Tiemann, J. Williams, and A. Yaw. We appreciate the dedication of K. Miranda, who assisted with both fieldwork and data entry while also working on an undergraduate honors thesis at Louisiana State University. KMR and LAR acquired the bulk of funding and DLB

obtained additional grants; DLB collected the data; DLB and KMR conceived and conducted the analyses, DLB drafted the paper with edits from KMR and LAR.

## Author Contributions

**Conceptualization:** Kevin M. Ringelman, Larry A. Reynolds.

**Formal analysis:** Dylan L. Bakner, Kevin M. Ringelman.

**Funding acquisition:** Kevin M. Ringelman, Larry A. Reynolds.

**Investigation:** Dylan L. Bakner, Larry A. Reynolds.

**Methodology:** Dylan L. Bakner, Kevin M. Ringelman, Larry A. Reynolds.

**Project administration:** Dylan L. Bakner, Kevin M. Ringelman, Larry A. Reynolds.

**Software:** Dylan L. Bakner.

**Supervision:** Kevin M. Ringelman.

**Validation:** Dylan L. Bakner.

**Visualization:** Dylan L. Bakner.

**Writing – original draft:** Dylan L. Bakner, Kevin M. Ringelman, Larry A. Reynolds.

**Writing – review & editing:** Dylan L. Bakner, Kevin M. Ringelman, Larry A. Reynolds.

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
