## [Decision Letter · Decision Letter 0]

11 Apr 2024

PONE-D-24-06740Wood duck nest survival and duckling recruitment is unaffected by interspecific brood parasitism from hooded mergansers and black-bellied whistling ducksPLOS ONE

Dear Dr. Bakner,

Thank you for submitting your manuscript to PLOS ONE. After careful consideration, we feel that it has merit but does not fully meet PLOS ONE’s publication criteria as it currently stands. Therefore, we invite you to submit a revised version of the manuscript that addresses the points raised during the review process.

We look forward to receiving your revised manuscript.

Kind regards,

Steven E. Travis, PhD

Academic Editor

PLOS ONE

Journal Requirements:

   "Funding for this project was provided by the Louisiana Department of Wildlife and Fisheries, United States Department of Agriculture, Nemours Wildlife Foundation, National Institute of Food and Agriculture McIntire-Stennis grant LAB94294, Louisiana State University College of Agriculture, Louisiana State University Agricultural Center, United Waterfowlers of Florida, and the Louisiana Ornithological Society."

6. We note that Figure 1 in your submission contain map/satellite images which may be copyrighted. All PLOS content is published under the Creative Commons Attribution License (CC BY 4.0), which means that the manuscript, images, and Supporting Information files will be freely available online, and any third party is permitted to access, download, copy, distribute, and use these materials in any way, even commercially, with proper attribution. For these reasons, we cannot publish previously copyrighted maps or satellite images created using proprietary data, such as Google software (Google Maps, Street View, and Earth). For more information, see our copyright guidelines: http://journals.plos.org/plosone/s/licenses-and-copyright.

Additional Editor Comments:

As you will see, both reviewers were enthusiastic about the importance of this contribution to the literature on interspecific brood parasitism, particularly considering the tremendous amount of data presented and the robustness of the statistical analysis.  However, Reviewer 1 would like to see some additional effort put into improving your figures, and both reviewers agree, and I strongly concur, that your analytical methods could use some additional explanation for the benefit of the many readers who may find them unfamiliar.  On the other hand, I do not feel it is necessary for you to create a table presenting definitions of the various acronyms used throughout your paper, although you may certainly follow this suggestion if you find it useful.

Additional line-by-line suggestions from my own independent review and those of two anonymous reviewers appear below.  You should pay careful attention to each of these comments in crafting your revisions.

Academic Editor comments:

Lines 20-24:  The first sentence should do a better job of summarizing the overall theme of the paper, which is about more than just hooded mergansers parasitizing wood ducks.  Combining the first two sentences and reorganizing a bit to bring wood ducks to the top should do the trick.

Lines 33-34:  Qualify “recruits” as adults.

Lines 39-41:  Why weren’t abandoned “dump nests” excluded from analysis?

Lines 41-42:  The effects of parasitism on recruitment could only be evaluated relative to hooded mergansers, but not whistling ducks, correct?  This should be clarified.

Lines 44-45: “minimally affected” might be a better way of describing it.

Lines 59-60:  It would be worth naming these three taxa.

Line 74:  Here and throughout remember to italicize species names.

Line 113:  replace “and” with “an.”

Line 212:  replace “predicted” with “modeled.”

Line 216:  I don’t really think sharing these model formulas is necessary, but if they are then it should be clarified why a model with only categorical explanatory variables (clutch type and site) would need an intercept and slope.  Maybe citing a reference would help clear things up.  Also, while site as a random effect might well contribute to model error, many other unmeasured factors would further contribute to error, so it doesn’t seem appropriate to depict this effect in your model definition simply as error.

Line 226:  parentheses should only be placed around the year.

Line 239:  At time t?

Line 252:  Again, unless you can cite a reference that establishes this modeling approach, which includes an intercept, a slope, and an error term in place of an actual random effect, I don’t think including a formal model definition is very useful.

Line 295:  Again, why depict your random effect, brood, as the sole source of error in this model definition, when there would be many other sources of unmeasured error?  Is there a reference you could cite for this approach?

Lines 296-297:  It’s not clear to me why female ducklings (as opposed to quantity of female ducklings per brood) wouldn’t serve as the trials for this analysis.  Or is that what you’re actually trying to say?

Lines 303-304:  Using mock nest initiation values for 17 nests would seem to create the possibility of artificially reduced variance terms.  Can you cite a reference to back up this approach?  Otherwise, why not omit these nests from the analysis?  Or at least compare results generated with and without these nests?

Lines 305-306:  I.e., you converted all observations to z-scores.  Why was this necessary or useful?

Line 309:  What about error that went beyond the random effects?  Or are you calling all variation not explained by the fixed effects random effects?

Line 336:  Omit “in those.”

Lines 344-346:  These percentages would be more informative if they were the percent of eggs hatching from the total eggs representing each group rather than all groups.

Line 354:  replace the Greek letter mu with x-bar or simply “mean” here and throughout.  Also, clarify that these are total clutch sizes representing both wood duck and IBP eggs.

Line 356:  why are two means given for mixed merganser nests?

Line 366 and Fig 3 caption:  Why are these clutch sizes described as “predicted” and not simply as mean clutch sizes?

Line 372:  Clarify that this DSR estimate cuts across all clutch types combined.

Line 374:  You should remind the reader of what the fixed effects for the DSR model were.

Lines 376-379:  How was the significance of these effects evaluated?

Lines 388-394:  Again, these percentages would be more informative if calculated within rather than among nest types.

Line 396:  Add “date” after “nest initiation.”

Lines 396-398:  How is the reader to gauge whether these effects are significant?

Line 399:  Again, remind the reader what fixed effects were being evaluated in this case.

Line 405:  Omit “marked.”

Reviewers' comments:

Reviewer's Responses to Questions

**Comments to the Author**

1. Is the manuscript technically sound, and do the data support the conclusions?

Reviewer #1: Yes

Reviewer #2: Yes

2. Has the statistical analysis been performed appropriately and rigorously? 

Reviewer #1: I Don't Know

Reviewer #2: Yes

3. Have the authors made all data underlying the findings in their manuscript fully available?

Reviewer #1: Yes

Reviewer #2: Yes

4. Is the manuscript presented in an intelligible fashion and written in standard English?

Reviewer #1: Yes

Reviewer #2: Yes

5. Review Comments to the Author

Reviewer #1: Dylan Bakner: Wood Duck recruitment and heterospecific nest parasitism

This manuscript summarizes and carefully analyses a huge data set addressing the effects of brood parasitism on both hatching success and recruitment (no less!) of Wood Duck eggs at three localities in Louisiana. While brood parasitism, both by other Wood Ducks and by Hooded Mergansers, is well known, this is the first large scale study to address the effects of parasitism by Black-bellied Whistling Ducks. It is an extraordinary data set that seems thoroughly well analyzed, although I am not at all qualified to assess the details of the modern analyses used in this paper.

I have only a minor recommendation concerning the text, and that is to consider adding a table summarizing the many short definitions used throughout the text. While each was obvious enough when presented, the ensemble, throughout the manuscript, became hard for me to remember.

My serious recommendation for improving the manuscript is that the figures be extensively revised to better communicate. There is just too much information packed into them, with conflicts about axis, for them to work well. To me, the best rule about figures is that if they cannot be directly moved into a textbook, without being redone, then they are not working well.

In general, most of the data figures are composites, presenting several things in each panel, but many of the panels completely lack the axies needed to make them understandable. It seems to me that the authors need to generate a better description of the wood duck breeding phenology before they dive into the many interactions with their two parasites that need interpretation. Further, the presence of both interspecific and conspecific parasitic eggs clearly shows that a single female’s clutch is smaller than optimal, and this continues to remain puzzling, as far as I know. Surely some brief mention of that might help inspire others to consider that conundrum.

Figure 2 has no Y axis that is labeled! Readers needs real dates at several places on the X axis so they don’t have to figure out months and days. Also, the legend suggests that the distributions do not include clutches that were not parasitized. Is this really true? If so, why were those excluded? (As an aside, it is extremely difficult for a reviewer to be forced to be searching for legends in the text when the figures, (without legends!!), appear at the at the end of the manuscript.)

I just don’t understand why there are not separate frequency distributions for all categories of nests: Plotting the frequency of wood duck nests by season, presumably would show up as a bimodal distribution (early and late, as shown in Figure 5?), needing some explanation: adults and juveniles or first nesters and re-nesters, or something else? THEN, the fraction parasitized needs to be listed in this or still another figure. It is clear that there is little overlap in the timing of parasitism by mergansers and tree ducks, but the grand phenology is poorly presented! And no effort is made to interpret the extreme difference in the laying phenology of the two parasites. I guess it seems to me that there needs to be a better descriptive presentation of the nesting phenology of wood ducks and their parasites before the strikingly bimodal temporals distributions of the two parasites is presented.

Figure 3 needs a clear explanation that this is a model and not actual averages. Further, it is nowhere in the legend that essentially all the merganser parasitized clutches were early, and the opposite for all the whistler parasitized clutches. So some reference is needed to Fig 2 to make this clear, as there is substantial overlap. Early and Late seem to refer to only Wood duck nest that were not parasitized in those time periods. But then, what becomes of late parasitic eggs laid by hoodeds, and vice versa for whistling ducks. These show up clearly in Figure 2!

Figure 4. Nothing in the legend reminds people that here, clutch size means the TOTAL number of parasitic eggs, as well as the number of host and parasitic wood duck eggs.

Figure 5. It is just unacceptable to have a distribution representing the number of observed nests that has no axis label! Further, it is unclear whether the X axis that would apply to the distribution of observed nests would be the same as that for nest initiation dates. In essence there are three figures that are stacked, presumably all with the same X, but that is not specified. Further the top two are different. One just a box plot of probabilities, the other a frequency distribution that may also apply to Fig. 2.

This is a stunning data set, and is making a great contribution to our understanding of how these two parasites are essentially having little effect on wood duck production in the southeast. But all that field work must deserve a bit of extra effort in revising the figures to make them communicate more effectively!

Sievert Rohwer

Reviewer #2: Review of PONE-D-24-06740

Wood duck nest survival and duckling recruitment is unaffected by interspecific brood parasitism from hooded mergansers and black-bellied whistling ducks.

General Comments:

I am a bit slow on this review, so I'll keep this short, not only in the interest of time, but also because I don't think an extensive review is needed. Overall, this is a solid manuscript that presents some useful information on the impacts of interspecific brood parasitism (IBP) by two species of cavity nesting waterfowl, Hooded Mergansers (HOME) and Black-bellied Whistling-ducks (BBWD), on a third species of cavity nesting waterfowl, the Wood Duck (WODU). There's considerable interest in the ecology, reproduction, and management of Wood Ducks, particularly in terms of nest and hatching success. Managers often use nest boxes to supplement natural cavities, and this has been an important management tool for decades. The potential utility of these practices can be impacted, however, by other species such as HOME and BBWD, who also use these nest boxes and lay their eggs in the nests of Wood Ducks, potentially reducing nest success. Accordingly, managers are concerned about these impacts. Moreover, Black-bellied Whistling-ducks have become increasingly abundant in the southern states as the authors note, and there's a potential for further expansion in the future. A basic study like this provides useful information on the potential impacts of such an expansion.

Beyond the management aspects, there's some interesting evolutionary ecology at play as well. Interspecific brood parasitism has been a focus of much research over the past several decades, particularly on the potential for coevolutionary arms races to arise between hosts and parasites. Most of that research has focused on altricial bird species, such as cuckoos and cowbirds and their hosts. Far fewer studies have examined interspecific parasitism in precocial species such as waterfowl. For precocial birds, the cost of parasitism may be considerably different, and there's a little evidence of co-evolutionary arms races, primarily because the impacts on the host may be much lower. Nonetheless, robust data on these impacts are limited, most of which has come from studies on two species - Canvasbacks and Redhead ducks, with smaller number of studies on Wood Ducks and Hooded Mergansers. Now, with Wood Ducks being exposed to two potential parasitic species, things get a bit more complicated. This is the first study to dive into this three-way interaction. The analyses are robust, and the manuscript is well written. In fact, as I read it, I wondered if the highly sophisticated analysis using Bayesian models were necessary for simple comparison of the effects of IBP on clutch size and hatching success, but they certainly take the level of rigor up a notch. The analysis of recruitment was a useful and novel aspect, although hampered by the small number of 50 returning ducklings (a surprisingly low recruitment rate) and the limited sample of recruits from mixed clutches. Nonetheless, the analysis of recruitment rates relative to initiation date was informative.

I have very few concerns or comments about the manuscript and make only a few small comments and suggestions below. Overall, this paper provides useful information, both from a management perspective and on the impacts of interspecific brood parasitism in precocial species of birds, for which there is a paucity of data.

Specific Comments:

Line 26, 30-31. This is an impressive number of nests, females, and ducklings! A huge effort on the part of the researchers and I just want to highlight the value of data collected over so many nests and over a period of four years. This alone makes the paper stand out among other studies with fewer samples and more limited study periods.

Line 80-82. You might also cite some of Charlotte Roy Nielsen's work on CBP in Wood Ducks - she has published several relevant papers and has done some great work that should be acknowledged.

Nielsen, C. R., P. G. Parker, and R. J. Gates. 2006. Intraspecific nest parasitism of cavity-nesting wood ducks: costs and benefits to hosts and parasites. Animal Behaviour 72:917–926.

Roy Nielsen, C. L., R. J. Gates, and P. G. Parker. 2006. Intraspecific nest parasitism of wood ducks in natural cavities: comparisons with nest boxes. The Journal of Wildlife Management 70:835–843.

Roy Nielsen, C. L., P. G. Parker, and R. J. Gates. 2008. Partial clutch predation, dilution of predation risk, and the evolution of intraspecific nest parasitism. The Auk 125:679–686.

Line 149. I recognize the challenges of visiting so many nest boxes on a regular interval. A weekly check (~7d) is typical of these kinds of studies, but I do wonder whether that generates enough encounters to provide sufficient data for a full-on DSR survival analysis - effectively 2-3 nest checks (encounters) per site?

Lines 211-221. I am, admittedly, a statistical dinosaur, and I very much appreciate the rigor and contemporary approach to these analyses. Using Bayesian multilevel Poisson models to estimate the effects of clutch type (HOME, BBWD parasitism) on clutch size is undoubtedly powerful, but I do wonder whether this is absolutely necessary. Is it akin to using a sledge hammer to simply drive a nail? I realize the value and strengths of Bayesian analyses and they are the new standard, but it does seem like a very sophisticated analysis for what should be a quite simple comparison of clutch size as a function of clutch type. I don't mean this as a criticism, but as an old-timer, I do sense a growing tendency to use the newest techniques simply because they are the newest technique, rather than one that's absolutely necessary. I don’t' disagree with doing things correctly and with the best possible method - perhaps the authors might make a short statement about their overall statistical approach and why it was chosen?

Line 221. So, hatch success was only considered for successful nests, which makes sense because including failed nests would conflate both nest success and hatch success, and would have an excess of 0's, leading to a very zero-inflated distribution. The authors might want to point out that in effect, they are undertaking a hierarchical approach to: (1) first assess the effect on nest success. If the nest is successful, then (2) assess the factors influencing clutch size and hatch success, and (3) then, with the sample of recruits, assess duckling survival and the effects of initiation date (not parasitism) on duckling recruitment. This may not be necessary but it would help a reader follow the inherent hierarchical framework and make it explicit.

Line 260-262. How did you remove/exclude males? Did you simply just split the sample of ducklings in half or did you actually sex ducklings (possible to do, but difficult and time consuming).

You should also clarify whether your recruits include ducklings that were "recaptured" by RFID reads, rather than being caught on the nest. That was never made clear (i.e., the hidden recruits - we have observed these females comprising a large fraction of the recruits in our own wood duck studies).

Line 281-282. Why is the entry for the transition from "duckling" in state t to "seen as adult" in state t+1 "0"in the observation matrix? I am probably not understanding how this analysis is done; naively I would think that entry would be the observed probability of marked ducklings that were then caught as adults (Φd). If not, how the is Φd) estimated (lines 288-290)? Maybe a bit more description for readers unfamiliar with this method.

Line 301-304. I am not sure that using "nest initiation date as replacement variable to represent the temporal variation in brood composition" really accounts for the effects of parasitism. Certainly, there is temporal overlap in the timing of early Wood Duck nests and Hooded Merganser parasitism, and late Wood Duck nests and Black-bellied Whistling-duck parasitism, but this is confounded by other seasonal changes in brood habitat, etc. that may have a strong influence on both mixed and non-mixed clutches. The question is whether, given (or controlling for) those seasonal patterns, there an additional influence of parasitism? I understand that small sample sizes of recruits for mixed clutches prevent such a contrast, but I don’t think using "initiation date" is valid "replacement variable". You simply are assessing the effect of initiation date (which is useful) but can infer little about the effect of parasitism per se.

Line 391-394. The percentages reported here refer to the proportions of the 50 recaptured ducklings, but that's not relevant given that different numbers of ducklings were tagged from each type of clutch. More useful would be the number of recaptured ducklings from each clutch type relative to the number that were tagged for that clutch type. So, 6 of 186 (3.25%) ducklings returned from mixed HOME clutches, 30 of 1389 (2.16%) returned from early WODU nests, 1 of 224 (0.45%) from mixed BBWD clutches, and 13 of 666 (1.98%) from late WODU clutches. That better represents the small number of recruits and the similarity among clutch types. The proportion of the 50 is meaningless given the different numbers tagged.

Line 412-416. This is a reasonable explanation, but might the lower survival of mixed merganser clutches also be due to aggressive interference among merganser and wood duck females, or perhaps just the effects of large rounder eggs of mergansers impacting wood duck incubation or nest success? Moreover, if this pattern were simply due to earlier nests including more "dump nests that are never incubated", that should be true for non-mixed early clutches as well and so would not explain the lower DSR of mixed merganser clutches. I think a more discussion of the reduced success in mixed merganser clutches is warranted.

Line 418-420. Wasn't the second term of the quadratic relationship with clutch size negative (line 377), such that DSR declines at larger clutch sizes? That also seems to be apparent in Figure 4. It appears you were interpreting this as indicating an asymptote, but it could be (and appears to be) a declining function after 30-35 eggs (you did not show the predicted function for larger clutch sizes but it trends down). Other studies of wood ducks (e.g. Semel and Sherman) found a decline in hatch success when clutch sizes become extreme. You might want to be more cautious about suggesting that there was "no evidence that excessive clutch sizes diminished survival".

Line 424-425. Again, a bit of caution is warranted here. Given that DSR is lower in the mixed merganser clutches, I would be cautious about suggesting the parasitic egg laying and larger clutch sizes can be a protective mechanism against nest abandonment.

Line 430-432. This is an interesting idea that over 25 years there has been adaptation to excessive brood parasitism. It's not clear how this might arise and without any evidence from other studies to support such a mechanism, perhaps this is too speculative? Admittedly, I am one who loves to speculate rampantly (and am often criticized for it), but I think you need a bit more evidence or at least a plausible mechanism for there to be support for this suggestion.

Overall, I enjoyed reading this manuscript and believe it is a very useful contribution. The sample sizes are extensive, the analyses are thorough and robust (even given my comments about using more sophisticate analytical methods than might be necessary), and there is a need for more studies such as this on precocial species if we are to fully understand the ecological, evolutionary, and management implications of IBP in birds.

I hope my comments are useful.

John Eadie

UC Davis

6. PLOS authors have the option to publish the peer review history of their article (what does this mean?). If published, this will include your full peer review and any attached files.

Reviewer #1: No

Reviewer #2: No

---

## [Author Response · Author response to Decision Letter 0]

24 May 2024

-Author’s responses to comments shown in red text.

Editor Comments

As you will see, both reviewers were enthusiastic about the importance of this contribution to the literature on interspecific brood parasitism, particularly considering the tremendous amount of data presented and the robustness of the statistical analysis. However, Reviewer 1 would like to see some additional effort put into improving your figures, and both reviewers agree, and I strongly concur, that your analytical methods could use some additional explanation for the benefit of the many readers who may find them unfamiliar. On the other hand, I do not feel it is necessary for you to create a table presenting definitions of the various acronyms used throughout your paper, although you may certainly follow this suggestion if you find it useful.

-Thanks! We hope the analytics in the revised manuscript are clearer. All the figures have been reworked according to Review 1. We also condensed the methods for concision as we agree we probably did go a bit overboard describing some rather simple glms.

ABSTRACT

Lines 20-24: The first sentence should do a better job of summarizing the overall theme of the paper, which is about more than just hooded mergansers parasitizing wood ducks. Combining the first two sentences and reorganizing a bit to bring wood ducks to the top should do the trick.

-Done.

Lines 33-34: Qualify “recruits” as adults.

-Done.

Lines 39-41: Why weren’t abandoned “dump nests” excluded from analysis?

-We included all samples in the analysis because nest survival estimates consider both successful and failed nests, regardless of the cause of failure. Abandonment is simply just a source of nest failure.

Lines 41-42: The effects of parasitism on recruitment could only be evaluated relative to hooded mergansers, but not whistling ducks, correct? This should be clarified.

-Our recruitment model was:

logit(pi) = α + β1 * nest initiation datei + β2 * brood sizei + εi,

So, we did not actually include parasite type (i.e., species that laid parasitic eggs) in this model due to low samples for recruits:

We recaptured 50 ducklings as adults (12.0%) hatched from clutches parasitized by hooded mergansers, 1 (2.0%) from a clutch parasitized by a whistling-duck, and 43 (86.0%) from clutches containing only wood duck eggs. 

So, as an alternative, we included nest initiation date and brood size as model covariates as they are linked to IBP involving both hooded mergansers and black-bellied whistling-ducks. We deleted the first part of this sentence and moved that second half to the line above where the recruitment results were first presented. Hopefully this clarifies things.

Lines 44-45: “minimally affected” might be a better way of describing it.

-Done. Also, we made this same change in the title.

INTRODUCTION

Lines 59-60: It would be worth naming these three taxa.

-Done. Added “(i.e., Anseriformes).” 

Line 74: Here and throughout remember to italicize species names.

-Done.

Line 113: replace “and” with “an.”

-Thanks for catching that.

METHODS

Line 212: replace “predicted” with “modeled.”

-Done.

Line 216: I don’t really think sharing these model formulas is necessary, but if they are then it should be clarified why a model with only categorical explanatory variables (clutch type and site) would need an intercept and slope. Maybe citing a reference would help clear things up. Also, while site as a random effect might well contribute to model error, many other unmeasured factors would further contribute to error, so it doesn’t seem appropriate to depict this effect in your model definition simply as error.

-Done, we removed the equation for this rather simple model. Valid claim, furthermore, we aren’t really interested in anything about the random effect from this model like others presented in this paper. The goal of this analysis was to simply derive estimates of “1) clutch size, 2) the number of wood duck eggs, and 3) the number of hatched wood duck eggs.” We will remove the random effect from this model as well as the other Poison model in this paper.

Line 226: parentheses should only be placed around the year.

-Thanks for catching that! 

Line 239: At time t?

-That’s correct. This has been corrected in the manuscript.

Line 252: Again, unless you can cite a reference that establishes this modeling approach, which includes an intercept, a slope, and an error term in place of an actual random effect, I don’t think including a formal model definition is very useful.

-This was removed. See the comment above (Line 216 comment).

Line 295: Again, why depict your random effect, brood, as the sole source of error in this model definition, when there would be many other sources of unmeasured error? Is there a reference you could cite for this approach?

-While depicted as the sole source of error, clearly that is not the case (as you suggested). In the paper we cited the seventh equation of Grosbois et al. (2008) which uses the variance estimates from a full fixed effect model and a separate random effect only model to calculate the variance explained by fixed effects. Now, why did we select these as random effects? We tried to pick the random effects that would explain the most variation in DSR and recruitment, which was based on our personal experiences being out in the field. 

Lines 296-297: It’s not clear to me why female ducklings (as opposed to quantity of female ducklings per brood) wouldn’t serve as the trials for this analysis. Or is that what you’re actually trying to say?

-Replaced “in” with “from” and “treated” with “considered” so this part of the manuscript reads “we considered the quantity of female wood duck ducklings from brood i as the number of trials (ni).” Hopefully that clears things up.

Lines 303-304: Using mock nest initiation values for 17 nests would seem to create the possibility of artificially reduced variance terms. Can you cite a reference to back up this approach? Otherwise, why not omit these nests from the analysis? Or at least compare results generated with and without these nests?

-Using means to consider data with missing covariate values has been used by previous researchers and we cite the methods described by Kéry and Royle (2016). Additionally, we reran the analysis to make sure this did influence our results. The model with these nests excluded yielded the following beta estimates:

-0.2092 (CI = -0.5546, 0.1222)

-0.7354 (CI = -1.1209, -0.3470),

and the variance explained was 18.7%. These results are the same as what we originally reported; therefore, we left this information unchanged in the manuscript. 

Lines 305-306: I.e., you converted all observations to z-scores. Why was this necessary or useful?

-Scaling puts all the model covariates on the same scale. This prevents obtaining extreme beta estimates when you have covariates that are on extremely different scales. 

Line 309: What about error that went beyond the random effects? Or are you calling all variation not explained by the fixed effects random effects?

-That is correct. Our fixed effects will explain some of the variation that we observed in different parameter estimates (i.e., explain some of the variation in nest survival and recruitment); however, there will still be more variation that is unexplained (i.e., things we failed to account for in our models). For example, in the recruitment model, our fixed effects explain ~19% of the variation in recruitment probability. That means there’s another 81% that’s explained by things not accounted for in our models.

RESULTS

Line 336: Omit “in those.”

-Done.

Lines 344-346: These percentages would be more informative if they were the percent of eggs hatching from the total eggs representing each group rather than all groups.

-Done. This section now reads: “We observed a total of 16,454 wood duck eggs, with 1,440 (8.8%) being laid in mixed merganser clutches, 8,777 (53.3%) in early clutches, 1,585 (9.6%) in mixed whistling-duck clutches, and 4,652 (28.3%) in late clutches. A total of 418 (29.0%) wood duck eggs hatched from mixed merganser clutches, 3,578 (40.8%) from early clutches, 607 (38.3%) from mixed whistling-duck clutches, and 1,734 (37.3%) from late clutches.”

Line 354: replace the Greek letter mu with x-bar or simply “mean” here and throughout. Also, clarify that these are total clutch sizes representing both wood duck and IBP eggs.

-Reworked this paragraph to include beta estimates to help it read cleaner; therefore, we removed mean estimates and replaced them with beta estimates. Then, Figure 3 does the job of showing model predictions. This paragraph now reads: “All our models achieved convergence according to trace plot inspections and "R" ^ values. Considering early clutches as the reference group, mixed merganser clutches were larger (β = 0.15, CI = 0.09, 0.20), late clutches were smaller (β = -0.27, CI = -0.30, -0.23), and mixed whistling duck clutches were comparable (β = 0.02, CI = -0.03, 0.07; Fig 3). There were fewer wood duck eggs in mixed merganser clutches (β = -0.11, CI = -0.18, -0.05), late clutches (β = -0.26, CI = -0.29, -0.22), and mixed whistling duck clutches (β = -0.31, CI = -0.37, -0.26; Fig 3). Likewise, fewer wood duck eggs hatched from mixed merganser clutches (β = -0.32, CI = -0.43, -0.21), late clutches (β = -0.26, CI = -0.32, -0.20), and mixed whistling duck clutches (β = -0.47, CI = -0.56, -0.39; Fig 3).”

Line 356: why are two means given for mixed merganser nests?

-That was a typo on our end. The correct value was 12.8 and is shown in Figure 3.

Line 366 and Fig 3 caption: Why are these clutch sizes described as “predicted” and not simply as mean clutch sizes?

-Y-axis is now labeled “Means”

Line 372: Clarify that this DSR estimate cuts across all clutch types combined.

-Now reads “Baseline DSR from our fixed effects model was 0.981 (CI = 0.974, 0.987) which yielded a nest success estimate of 0.455 (CI = 0.340, 0.585).” Also added “baseline” to the following statement in the methods: “We used a logit link function to evaluate the relationship between covariates and DSR, where α is baseline DSR on the logit scale.” This states DSR is simply α.

Line 374: You should remind the reader of what the fixed effects for the DSR model were.

-This now reads: “Baseline DSR from our model containing clutch type and clutch size along with its quadratic form as fixed effects was 0.981 (CI = 0.974, 0.987), which yielded a nest success estimate of 0.455 (CI = 0.340, 0.585).”Also, we add “For models containing clutch type as a covariate, normal clutches were used as the reference group.” To the end of the method section to hopefully make things easier to interpret. 

Lines 376-379: How was the significance of these effects evaluated?

-See second to last sentence in the last paragraph of methods. “From our analyses, we report posterior means and coefficient values (β), where credible intervals (after, CI) that did not overlap zero indicated significance effects from model coefficients at the 0.05 alpha level.”

Lines 388-394: Again, these percentages would be more informative if calculated within rather than among nest types.

-Reviewer 2 had a similar comment. This now reads, “We recaptured 50 (2.0%) ducklings as breeding adults with 6 (3.3%) returning from mixed merganser clutches, 30 (2.2%) from early clutches, 1 (0.5%) from a mixed whistling-duck clutch, and 13 (1.2%) from late clutches.”

Line 396: Add “date” after “nest initiation.”

-Done.

Lines 396-398: How is the reader to gauge whether these effects are significant?

-See second to last sentence in the last paragraph of methods. “From our analyses, we report posterior means and coefficient values (β), where credible intervals (after, CI) that did not overlap zero indicated significance effects from model coefficients at the 0.05 alpha level.”

Line 399: Again, remind the reader what fixed effects were being evaluated in this case.

-Now reads: “The across-brood variance estimate was 0.544 (CI = 0.025, 1.352) for the model containing fixed effects (i.e., nest initiation date and brood size) and 0.671 (CI = 0.036, 1.527) for the random-effects-only model, suggesting fixed effects explained 18.8% of the across-brood variation in recruitment probability.”

Line 405: Omit “marked.”

-Done.

Review #1 Comments

This manuscript summarizes and carefully analyses a huge data set addressing the effects of brood parasitism on both hatching success and recruitment (no less!) of Wood Duck eggs at three localities in Louisiana. While brood parasitism, both by other Wood Ducks and by Hooded Mergansers, is well known, this is the first large scale study to address the effects of parasitism by Black-bellied Whistling Ducks. It is an extraordinary data set that seems thoroughly well analyzed, although I am not at all qualified to assess the details of the modern analyses used in this paper.

I have only a minor recommendation concerning the text, and that is to consider adding a table summarizing the many short definitions used throughout the text. While each was obvious enough when presented, the ensemble, throughout the manuscript, became hard for me to remember.

-We decided not to include a table as we reworked the manuscript substantially and hope that it is now clearer.

My serious recommendation for improving the manuscript is that the figures be extensively revised to better communicate. There is just too much information packed into them, with conflicts about axis, for them to work well. To me, the best rule about figures is that if they cannot be directly moved into a textbook, without being redone, then they are not working well.

-We greatly appreciate your comments on out figures and feel they are now much more impactful.

In general, most of the data figures are composites, presenting several things in each panel, but many of the panels completely lack the axies needed to make them understandable. It seems to me that the authors need to generate a better description of the wood duck breeding phenology before they dive into the many interactions with their two parasites that need interpretation. Further, the presence of both interspecific and conspecific parasitic eggs clearly shows that a single female’s clutch is smaller than optimal, and this continues to remain puzzling, as far as I know. Surely some brief mention of that might help inspire others to consider that conundrum.

-We added axis labels where necessary. 

FIGURES

Figure 2 has no Y axis that is labeled! Readers needs real dates at several places on the X axis so they don’t have to figure out months and days. Also, the legend suggests that the distributions do not include clutches that were not parasitized. Is this really true? If so, why were those excluded? (As an aside, it is extremely difficult for a reviewer to be forced to be searching for legends in the text when the figures, (without legends!!), appear at the at the end of the manuscript.)

-We added the requested axis label, real dates rather than Julian dates, and also include distribution plots for early and late clutches. In this figure the distribution is derived using a probability density function. I agree that would be frustrating, but that is how PLOS ONE required us to submit. 

This figure captions now reads: “Fig 2. Density plots showing nest initiation dates for wood duck (Aix sponsa) clutches categorized into four groups: clutches containing parasitic hooded merganser eggs (Lophodytes cucullatus; Mixed HOME), early normal clutches (i.e., clutches with only wood duck eggs), clutches containing parasitic black-bellied whistling-duck eggs (Dendrocygna autumnalis; Mixed BBWD), and late normal clutches. The vertical dashed line represents April 1, which is used as the cutoff date for early and late wood duck nests.” 

I just don’t understand why there are not separate frequency distributions for all categories of nests: Plotting the frequency of wood duck nests by season, presumably would show up as a bimodal distribution (early and late, as shown in Figure 5?), needing some explanation: adults and juveniles or first nesters and re-nesters, or something else? THEN, the fr

---

## [Editor Report · Decision Letter 1]

30 May 2024

PONE-D-24-06740R1Wood duck nest survival and duckling recruitment is minimally affected by interspecific brood parasitism from hooded mergansers and black-bellied whistling-ducksPLOS ONE

Dear Dr. Bakner,

Thank you for submitting your manuscript to PLOS ONE. After careful consideration, we feel that it has merit but does not fully meet PLOS ONE’s publication criteria as it currently stands. Therefore, we invite you to submit a revised version of the manuscript that addresses the points raised during the review process.

 Your revised manuscript is much improved and satisfactorily addresses nearly all of the reviewers' comments on your original submission.  Just a few items remain to be cleared up. Lines 235-236 and line 243:  Considering that epsilon is typically used to symbolize unexplained error, as opposed to the variance explained by random effects, I would urge you to replace it with “site ID” in this model formula, and then to remove your parenthetical reference to epsilon on line 243.  This approach would also require removing the definition of epsilon on line 236.

Lines 289 and 293:  Following the same logic, replace the epsilon with “brood ID” on line 289 and delete the parenthetical epsilon on line 293.

Line 297:  I think you were better off with the language you had in place previously, although I would suggest adding a brief explanation of why the standardization was necessary, similar to what you provided in your response to reviewers.

Fig. 1:  Please limit this figure to three panels, one for mixed HOME, one for mixed BBWD, and one for non-IBP nests.  By separating the latter into early and late phases of the nesting period, you’re placing a time-constraint on each distribution that doesn’t naturally exist.  Creating one distribution for the non-parasitized nests will presumably create the sort of bimodal distribution Dr. Rohwer is alluding to in his comments, and even if it doesn’t it won’t invalidate your approach to dividing up the nesting season into two distinct periods for the purposes of statistical analysis.

Fig. 4:  Please remove “Observed nests” from the figure and move “Recruited female ducklings” to the y-axis so that it reads “Density of recruited female ducklings.”  Also, clarify in the accompanying caption that these data consider females only.

We look forward to receiving your revised manuscript.

Kind regards,

Steven E. Travis, PhD

Academic Editor

PLOS ONE
---

## [Author Response · Author response to Decision Letter 1]

4 Jun 2024

-Author’s responses to comments shown in red text.

Editor Comments

ABSTRACT

INTRODUCTION

METHODS

Lines 235-236 and line 243: Considering that epsilon is typically used to symbolize unexplained error, as opposed to the variance explained by random effects, I would urge you to replace it with “site ID” in this model formula, and then to remove your parenthetical reference to epsilon on line 243. This approach would also require removing the definition of epsilon on line 236.

-Done!

Lines 289 and 293: Following the same logic, replace the epsilon with “brood ID” on line 289 and delete the parenthetical epsilon on line 293.

-Done!

Line 297: I think you were better off with the language you had in place previously, although I would suggest adding a brief explanation of why the standardization was necessary, similar to what you provided in your response to reviewers.

-Done! This now reads: “Prior to running our models, we scaled all variables to have a mean of 0 and a standard deviation of 1. This process places all covariates on the same scale, facilitating the comparison of beta estimates across different covariates [54].”

RESULTS

DISCUSSION

FIGURES

Fig. 1: Please limit this figure to three panels, one for mixed HOME, one for mixed BBWD, and one for non-IBP nests. By separating the latter into early and late phases of the nesting period, you’re placing a time-constraint on each distribution that doesn’t naturally exist. Creating one distribution for the non-parasitized nests will presumably create the sort of bimodal distribution Dr. Rohwer is alluding to in his comments, and even if it doesn’t it won’t invalidate your approach to dividing up the nesting season into two distinct periods for the purposes of statistical analysis.

-Done!

Fig. 4: Please remove “Observed nests” from the figure and move “Recruited female ducklings” to the y-axis so that it reads “Density of recruited female ducklings.” Also, clarify in the accompanying caption that these data consider females only.

-Done! Given the distribution of nest initiation dates for all the nests we observed throughout the study, which was moved to Fig 1 based on the previous comment, we deleted the top plot from this figure and left everything else the same. We are happy to modify this figure

---

## [Editor Report · Decision Letter 2]

7 Jun 2024

Wood duck nest survival and duckling recruitment is minimally affected by interspecific brood parasitism from hooded mergansers and black-bellied whistling-ducks

PONE-D-24-06740R2

Dear Dr. Bakner,

We’re pleased to inform you that your manuscript has been judged scientifically suitable for publication and will be formally accepted for publication once it meets all outstanding technical requirements.

Kind regards,

Steven E. Travis, PhD

Academic Editor

PLOS ONE
---

## [Editor Report · Acceptance letter]

21 Jun 2024

PONE-D-24-06740R2 

PLOS ONE

Dear Dr. Bakner, 

I'm pleased to inform you that your manuscript has been deemed suitable for publication in PLOS ONE. Congratulations! Your manuscript is now being handed over to our production team.

Kind regards, 

on behalf of

Dr. Steven E. Travis 

Academic Editor

PLOS ONE